# Spent Grain from Malt Whisky: Assessment of the Phenolic Compounds

**DOI:** 10.3390/molecules26113236

**Published:** 2021-05-27

**Authors:** Ancuța Chetrariu, Adriana Dabija

**Affiliations:** Faculty of Food Engineering, Stefan cel Mare University of Suceava, 720229 Suceava, Romania; ancuta.chetrariu@fia.usv.ro

**Keywords:** spent grain, phenolic compounds, ultrasound-assisted, high stirring rate

## Abstract

In order to extract antioxidant phenolic compounds from spent grain (SG) two extraction methods were studied: the ultrasound-assisted method (US) and the Ultra-Turrax method (high stirring rate) (UT). Liquid to solid ratios, solvent concentration, time, and temperature/stirring rate were optimized. Spent grain extracts were analyzed for their total phenol content (TPC) (0.62 to 1.76 mg GAE/g SG DW for Ultra-Turrax pretreatment, and 0.57 to 2.11 mg GAE/g SG DW for ultrasound-assisted pretreatment), total flavonoid content (TFC) (0.6 to 1.67 mg QE/g SG DW for UT, and 0.5 to 1.63 mg QE/g SG DW for US), and antioxidant activity was measured using 2,2-diphenyl-2-picrylhydrazyl (DPPH) free radical (25.88% to 79.58% for UT, and 27.49% to 78.30% for UT). TPC was greater at a high stirring rate and high exposure time up to a certain extent for the Ultra-Turrax method, and at a high temperature for the ultrasound-assisted method. P-coumaric acid (20.4 ± 1.72 mg/100 SG DW for UT, and 14.0 ± 1.14 mg/100 SG DW for US) accounted for the majority of the phenolic found compounds, followed by rosmarinic (6.5 ± 0.96 mg/100 SG DW for UT, and 4.0 ± 0.76 mg/100 SG DW for US), chlorogenic (5.4 ± 1.1 mg/100 SG DW for UT, and non-detectable for US), and vanillic acids (3.1 ± 0.8 mg/100 SG DW for UT, and 10.0 ± 1.03 mg/100 SG DW for US) were found in lower quantities. Protocatechuic (0.7 ± 0.05 mg/100 SG DW for UT, and non-detectable for US), 4-hydroxy benzoic (1.1 ± 0.06 mg/100 SG DW for UT, and non-detectable for US), and caffeic acids (0.7 ± 0.03 mg/100 SG DW for UT, and non-detectable for US) were present in very small amounts. Ultrasound-assisted and Ultra-Turrax pretreatments were demonstrated to be efficient methods to recover these value-added compounds.

## 1. Introduction

Whisky is one of the most popular alcoholic beverages made from cereals, and generally are all produced in a similar manner [1,2]. Today, whisky is drunk in over 200 countries around the world, making it one of the most consumed alcoholic beverages globally [3,4]. Whisky consumption per capita varies by country, with France at the top by about 2 L per person per year, while India is the country with the highest total consumption [5]. There are currently three types of whisky: malt whisky, grain whisky and blended whisky. Worldwide, whisky is subject to strict regulations, with many types and classes regarding the fermentation of grains, distillation and aging in wooden barrels [3,6,7].

In Europe, whisky is defined as an alcoholic beverage produced by distillation of malt or cereal wort. There are many types of whisky depending on the country in which it is produced. In European countries, whisky is mainly produced with malt made from barley, water and cereals, such as wheat or rye. However, depending on the region, it can be obtained by different production methods [1,2]. The main production countries are Scotland, the USA, Ireland, Canada, and Japan. The top 10 whisky markets by volume are India, the USA, France, Japan, the United Kingdom, Spain, Canada, Brazil, South Africa, and Germany [8,9].

In 2017, the Alexandrion Group became the first Romanian company that owns in its portfolio a single malt whisky produced 100% in Romania. The decision to produce a single malt in Romania was a strategic one, in line with the company’s values as a leader in the local liquor market.

The management of by-products and residues is considered a significant economic and environmental problem in whisky production. The main by-products resulting from the production of malt whisky are spent grain or draff, the starch-depleted waste grains that remain after the mashing stage, pot ale, liquid residues from the first stage of distillation, and the spent yeast resulting from distillation (Figure 1) [10,11].

By-products are clearly defined by European legislation in accordance with Article 5 of Directive 2008/98/EC (EU 2008) [12]. The annual amount of spent grain or draff resulting from the obtained malt whisky is ~2400 tons per million liters of produced alcohol [13]. The high humidity content of 74–80% of this by-product gives it a short shelf life; being prone to microbial contamination, it has limited use and is not suitable for long-distance transport [14]. Spent grain contains peel, pericarp, and seeds with high cellulose levels (16.8–25.4%), hemicellulose (mainly arabinoxylans) (21.8–28.4%), lignin (11.9–27.8%), protein (20–25%), or fibers [4,10]. Most of the weight of spent grain is water, distributed during the mashing process.

The possibilities of the recovery of spent grains (draff) are presented in the literature. The traditional method is the use of these by-products, either in wet or dry form, as feed (mainly for cattle or sheep), due to both its nutritional content and low cost [15,16,17,18].

The valorization of spent grains is of great interest to obtain different high-value biocompounds that can be extracted, purified, and reused as functional ingredients in the food industry and in other industries. Spent grains are a potentially valuable source of phenolic compounds, including phenolic acids (benzoic and cinnamic acid derivatives), flavonoids, proanthocyanidins, tannins, and amino phenolic compounds, and they are found in both the free and the bound form [19,20]. Phenolic compounds are currently the focus of much attention due to their important antioxidant and antiradical properties [21,22]. A few studies have investigated brewer’s spent grain (BSG) as a source of bioactive phenolic compounds such as hydroxycinnamic acids, especially ferulic and p-coumaric acids [23,24,25].

For the isolation of phenolic compounds from spent grains, the first step is extraction. For example, a number of extraction methods have been applied to BSG: microwave-assisted extraction (MAE) [21,22], ultrasound-assisted extraction (UAE) [19,20], microwave energy treatment (MWT) [24], pulsed electric field (PEF) [26], supercritical fluids extraction, and high-pressure processes [27]. These methods use a solvent such as methanol, ethanol, acetone [28], ethyl acetate, a mixture of acetone and water at 50% (*v*/*v*), saponification with NaOH solution [29], chemical/enzymatic hydrolysis [20], etc. Patrignani et al. proposed the use of water as an alternative for the recovery of phenolic compounds from BSG [24]. It is necessary to optimize the extraction conditions to maximize phenolic compound recovery: the type of solvent and its concentration, the solvent/solid ratio, the number of extraction steps, pH, time of contact, temperature, etc. [25].

Meneses et al. used different solvents at different concentrations in their study to extract phenolic compounds from brewer’s spent grain. When water was used, the total phenolic compounds were 3.59 ± 0.46 mg GAE/g BSG, the results for hexane were 4.44 ± 0.44 mg GAE/g BSG, for ethyl acetate 2.14 ± 0.53 mg GAE/g BSG, for acetone in concentration between 20% and 100% *v*/*v* the results ranged from 5.37 ± 0.1 mg GAE/g BSG to 9.90 ± 0.41 mg GAE/g BSG, for ethanol in concentration between 20% and 100% *v*/*v,* results ranged from 4.26 ± 0.51 mg GAE/g BSG to 7.13 ± 0.24 mg GAE/g BSG, and for different concentrations of methanol (20–100% *v*/*v*), results ranged from 2.90 ± 0.43 mg GAE/g BSG to 6.46 ± 0.50 mg GAE/g BSG [29]. Carciochi et al. used an ethanol 70% aqueous mixture in different liquid-to-solid ratios (10, 20, 30 mL solvent/g dry BSG) with values of 2.59, 2.74, and 3.07 mg GAE/g dry BSG, respectively, for the total phenolic compounds [30].

There are several patents on by-product recycling for use as food for human consumption, e.g., using spent grain as an ingredient in the production of bakery products, meat products, sauces, nutraceuticals, milk drinks, textured soft drinks, non-alcoholic beverages and flavorings [31]. It is necessary to add further value to these by-products and to develop new low-cost recovery methods, to make them economically attractive to companies in the whisky industry.

The paper proposes the phenolic characterization of spent grain (draff) resulting from being the only producer in the whisky industry in Romania, with the aim of identifying ways of valuing it in the context of sustainable development.

## 2. Results

### 2.1. Characterization of Spent Grain

The composition of spent grain used in this study (%*w*/*w*, dry basis) was moisture (5.04 ± 0.42%), ash (3.47 ± 0.02%), protein (18.88 ± 0.37%), lipids (7.11 ± 0.39%), and fibers (22.67 ± 0.42%), which are in agreement with the results obtained for spent grain from malt whisky [13,32]. Akunna & Walker [13] reported ash (5%), protein (15%), and lipids (10%), and Barker [32] reported ash (2.8%), protein (23.6%), and fiber (17%). del Rio et al. [32] found the next result for ash (4.9 ± 0.1% dry weight), protein (14.5 ± 1.0% dry weight), and lipids (9.2 ± 0.2% dry weight) for brewer’s spent grain.

### 2.2. Model Fitting and Statistical Analysis

A Box-Behnken (BB) design was used, which has four independent variables, each with three levels and three replications at the center point. SG samples were extracted by means of the procedure described in Section 3.5. Table 1 presents the experimental values used for total polyphenol content (TPC), total flavonoid content (TFC), and scavenging DPPH free radical activity (DPPH) for both extractions.

Analysis of variance (ANOVA) validates the model, which fits the observed variation in extracted protein at the designed extraction level. The model fits and adequately explains the observed variation if the F-test is significant at the 5% level (*p* < 0.05). ANOVA for the predicted model for this experiment is given in Table 2.

#### 2.2.1. Optimization of Extraction Parameters and Validation of the Models

According to the results presented in Table 3, the optimal values for the Ultra-Turrax method were 0.849 g for the sample mass, 54% for the concentration of methanol used, and a stirring rate of 30, 000 rpm for 90s. The extraction under optimum conditions reached 1.37 ± 0.11 mg GAE/g dry spent grain of TPC, 1.34 ± 0.15 mg QE/g dry spent grain of TFC, and 54.77 ± 3.75% inhibition of DPPH. The optimal values for the ultrasound-assisted method were 0.875 g for the sample mass, 78% for methanol concentration, at a temperature of 70° Celsius for 22min. The following results were obtained under these conditions: 1.42 ± 0.15 mg GAE/g dry spent grain of TPC, 1.25 ± 0.11 mg QE/g dry spent grain of TFC, and 74.20 ± 4.39% inhibition of DPPH.

The predicted results for the responses were verified, and differences ≤5% were obtained between the predicted and verified values.

BSG contains a high amount of lignin that is connected to its cell wall polysaccharides by phenolic acids [29], and the results obtained by conventional methods are lower than the results obtained by ultrasonic and Ultra-Turrax extraction methods. The total phenol content for extraction in the water bath is 1.32 ± 0.25 mg GAE/g SG DW, and 1.12 ± 0.19 mg GAE/g SG DW for the maceration method. The total flavonoid content for water bath extraction is 0.83 ± 0.03 mg GAE/g SG DW, and 0.78 ± 0.04 mg GAE/g SG DW for the maceration method. Radical scavenging activity for extraction in the water bath is 57.89 ± 1.15 mg GAE/g SG DW, and 53.84 ± 1.35 mg GAE/g SG DW for the maceration method.

#### 2.2.2. Total Phenolic Compounds

The BB design was employed to study the interactions between four individual parameters. The phenolic compound extraction rate for Ultra-Turrax pre-treatment ranged from 0.62 to 1.76 mg GAE/g SG DW. In comparison, for ultrasound-assisted pre-treatment, the phenolic compound extraction rate varied from 0.57 to 2.11 mg GAE/g SG DW. Those results are in agreement with other studies [33,34,35,36,37,38]. Multiple regression analysis was used to analyze the experimental data, and thus a quadratic polynomial equation of the response surfaces was obtained as follows (*** *p* < 0.001; ** *p* < 0.01; * *p* < 0.05):TPC Ultra-Turrax = 1.40 + 0.22 * X_1_*** − 0.05 * X_2_ − 0.20 * X_3_*** + 0.22 * X_4_*** − 0.01 * X_1_X_2_ + 0.05 * X_1_X_3_ − 0.01 * X_1_X_4_ + 0.03 * X_2_X_3_ − 0.04 * X_2_X_4_ − 0.004 * X_3_X_4_ + 0.004 * X_1_^2^ + 0.05 * X_2_^2^ − 0.24 * X_3_^2^** − 0.08 * X_4_^2^(1)

One factor, namely, solvent concentration (X_3_), had a significant negative effect on the phenolic content (*p* < 0.05), while the stirring rate and time (X_1_ and X_4_) had a significant positive effect (*p* < 0.05). From Equation (1), it can be observed that the quadratic term of solvent concentration (X_3_^2^) had the highest negative effect of phenolic content in Ultra-Turrax extraction.
TPC Ultrasound-assisted =1.44 − 0.32 * X_1_*** + 0.07 * X_2_ + 0.12 * X_3_* + 0.25 * X_4_*** + 0.11 * X_1_X_2_ + 0.27 * X_1_X_3_* − 0.001 * X_1_X_4_ − 0.19 * X_2_X_3_ − 0.02 * X_2_X_4_ − 0.16 * X_3_X_4_ − 0.06 * X_1_^2^ − 0.18 * X_2_^2*^ − 0.19 * X_3_^2^* + 0.10 * X_4_^2^(2)

Sample mass (X_1_) had a significant negative effect on phenolic content (*p* < 0.05), and time and temperature factors (X_3_ and X_4_) had a significant positive effect (*p* < 0.05). It can be observed from Equation (2) that the interaction between sample mass and time had (X_1_ and X_3_) the highest positive effect, while the quadratic term of solvent concentration (X_2_) and time (X_3_) had the highest negative effect. Other studies have shown that temperature plays an important role in the extraction of phenols; the higher the temperature, the higher the solubility of the compounds and the diffusion coefficient [33,36]. Gupta et al. [39] determined a maximum TPC when the solid-to-liquid ratio was the lowest in brewing by-products, which is in agreement with Pedro et al. [40].

The analysis of variance showed that the quadratic regression model is statistically significant (*p* < 0.0001) in both cases. In Figure 2 the 3D evolution of phenolic content is presented in the function of the parameters employed. For Ultra-Turrax, the pre-treatment coefficient of regression R^2^ is 0.9173, and the adjusted R^2^ is 0.8208, and for the ultrasound-assisted pre-treatment coefficient of regression R^2^ 0.8882, and the adjusted R^2^ is 0.7578.

From Figure 2, the lowest phenolic content from spent grain was recorded at the correlation of the following parameter values: 0.9 g of sample, 60% solvent concentration, 10 min and 50 °C, and the highest content was observed at the correlation between 0.3 g of the sample, 60% solvent concentration, 20 min, and 70 °C for treatment assisted by ultrasound.

Spent grain subjected to high homogenization released the lowest phenolic content, which was recorded at the correlation of the following parameter values: 0.6 g of sample, 80% solvent concentration, 60 s and 10,000 rpm, and the highest content was observed at the correlation between 0.3 g of sample, 60% solvent concentration, 60 s, and 30,000 rpm. The increase of TPC with stirring rate and time increase showed that a good homogenization helps to release phenolic compounds. Ultra-Turrax pre-treatment by aqueous methanol extraction provides a slightly better yield of phenolic compounds compared to ultrasound-assisted extraction. The origin of the raw material, extraction conditions and previous processing clearly influence the TPC content [37].

#### 2.2.3. Total Flavonoid Content

The flavonoid compound extraction rate for Ultra-Turrax pre-treatment ranged from 0.60 to 1.67 mg QE/g SG DW. In comparison, for ultrasound-assisted pre-treatment, the flavonoid compound extraction rate varied from 0.50 to 1.63 mg QE/g SG DW, which is in agreement with other studies [38,41,42,43]. Multiple regression analysis was used to analyze the experimental data, and thus a quadratic polynomial equation of the obtained response surfaces is as follows:TFC Ultra-Turrax = 1.36 + 0.12 * X_1_** + 0.10 * X_2_* − 0.11 * X_3_* + 0.28 * X_4_*** + 0.002 * X_1_X_2_ + 0.21 * X_1_X_3_ − 0.09 * X_1_X_4_** + 0.08 * X_2_X_3_ − 0.06 * X_2_X_4_ − 0.007 * X_3_X_4_ + 0.05 * X_1_^2^ − 0.09 * X_2_^2^ − 0.19 * X_3_^2^** − 0.12 * X_4_^2^(3)

One factor, namely, solvent concentration (X_3_), has a significant negative effect on the flavonoid content (*p* < 0.05), while stirring rate, liquid to solid ratio, and time (X_1,_ X_2,_ and X_4_) have a significant positive effect (*p* < 0.05). From Equation (3), it can be observed that the interaction of stirring rate and time (X_1_X_4_), and the quadratic term of solvent concentration (X_3_^2^) have the highest negative effect.
TFC Ultrasound-assisted = 1.28 − 0.02 * X_1_ − 0.14 * X_2_** − 0.02 * X_3_ + 0.04 * X_4_ − 0.20 * X_1_X_2_* + 0.23 * X_1_X_3_^**^ − 0.15 * X_1_X_4_* + 0.22 * X_2_X_3_* − 0.03 * X_2_X_4_ − 0.47 * X_3_X_4_*** + 0.03 * X_1_^2^ − 0.07 * X_2_^2^ − 0.32 * X_3_^2^*** + 0.03 * X_4_^2^(4)

Solvent concentration has a significant negative effect on the flavonoid content (*p* < 0.05), while the other factors have non-significant effects. From Equation (4), it can be observed that the interaction of liquid-to-solid ratio and time (X_1_X_3_), and the interaction between solvent concentration and time (X_2_X_3_) have the highest positive effect, while the interaction of time and temperature (X_3_X_4_), the interaction of liquid-to-solid ratio and temperature (X_1_X_4_), and the quadratic term of time (X_3_^2^) have the highest negative effect.

The analysis of variance has shown that the quadratic regression model is statistically significant (*p* < 0.0001) in both extractions. The 3D evolution of flavonoid content depending on the employed parameters is presented in Figure 3. For Ultra-Turrax, the pre-treatment coefficient of regression R^2^ is 0.9031, the adjusted R^2^ is 0.7900, and for the ultrasound-assisted pre-treatment coefficient of regression R^2^ 0.8994, the adjusted R^2^ is 0.7820.

From Figure 3, the lowest flavonoid content was recorded at the correlation of the following parameter values: 0.6 g of sample, 80% solvent concentration, 10 min and 50 °C, and the highest content was observed at the correlation between 0.9 g of sample, 40% solvent concentration, 20 min, and 50 °C for treatment assisted by ultrasound. In comparison to this, the lowest flavonoid content was recorded at the correlation of the following parameter values: 0.3 g of sample, 60% solvent concentration, 30 s and 20,000 rpm, and the highest content was observed at the correlation between 0.6 g of sample, 60% solvent concentration, 90 s, and 10,000 rpm for Ultra-Turrax pre-treatment.

#### 2.2.4. DPPH Assay of Scavenging Activity

Radical scavenging activity for Ultra-Turrax pre-treatment ranged from 25.88% to 79.58%. Ultrasound-assisted pre-treatment radical scavenging activity ranged from 27.49% to 78.30%, which is in agreement with other studies [33,38,39]. DPPH (2, 2-diphenyl-1-picrylhydrazyl) can be dissolved in organic solvents, especially in alcoholic solvents [44]; in this study methanol is the solvent used. The polynomial equations for DPPH are described below (Equations (5) and (6)):DPPH Ultra-Turrax = 54.75 + 8.42 * X_1_*** + 16.89 * X_2_*** + 1.86 * X_3_ + 6.45 * X_4_*** − 0.97 * X_1_X_2_ + 2.38 * X_1_X_3_ − 1.58 * X_1_X_4_ + 0.09 * X_2_X_3_ − 0.31 * X_2_X_4_ + 1.27 * X_3_X_4_ + 2.08 * X_1_^2^ − 4.41 * X_2_^2^ + 2.90 * X_3_^2^ − 0.51 * X_4_^2^(5)

Three factors, namely, stirring rate, liquid to solid ratio, and time (X_1,_ X_2,_ and X_4_) have a significant positive effect (*p* < 0.05) on antioxidant capacity.
DPPH Ultrasound assisted = 74.13 + 7.03 * X_1_*** + 5.92 * X_2_** + 9.13 * X_3_*** + 7.99 * X_4_*** + 12.19 * X_1_X_2_*** + 3.59 * X_1_X_3_ − 1.38 * X_1_X_4_ − 2.83 * X_2_X_3_ − 1.05 * X_2_X_4_ − 9.22 * X_3_X_4_** − 11.77 * X_1_^2^*** − 16.27 * X_2_^2^*** − 10.06 * X_3_^2^*** − 4.83 * X_4_^2^*(6)

All factors, namely, solvent concentration, stirring rate, liquid-to-solid ratio, and time (X_1,_ X_2,_ X3, and X_4_) had a significant positive effect (*p* < 0.05) on antioxidant capacity in ultrasound-assisted extraction. From Equation (6), it can be observed that the quadratic term of solvent concentration (X_2_^2^) had the highest negative effect, while the interaction of liquid-to-solid ratio and solvent concentration (X_1_X_2_) had the highest positive effect.

The analysis of variance has shown that the quadratic regression model is statistically significant (*p* < 0.001) in both extraction pre-treatments. The 3D evolution of the antioxidant activity content function of the parameters employed is presented in Figure 4. For Ultra-Turrax the pre-treatment coefficient of regression R^2^ is 0.9477, and the adjusted R^2^ is 0.8868, and for the ultrasound-assisted pre-treatment coefficient of regression R^2^ 0.9483, the adjusted R^2^ is 0.8881.

From Figure 4, the lowest antioxidant activity was recorded at the correlation of the following parameter values: 0.6 g of sample, 40% solvent concentration, 10 min and 50 °C, and the highest activity was observed at the correlation between 0.6 g of sample, 60% solvent concentration, 20 min, and 50 °C for treatment assisted by ultrasound. In comparison to this, the lowest antioxidant activity was recorded at the correlation of the following parameter values: 0.3 g of sample, 60% solvent concentration, 60 s and 10,000 rpm, and the highest activity was observed at the correlation between 0.9 g of sample, 80% solvent concentration, 60 s, and 20,000 rpm for Ultra-Turrax treatment.

### 2.3. HPLC Analysis of Phenolic Compounds

Qualitative and quantitative analysis of the phenolic extract obtained at the optimized conditions were performed using a high-performance liquid chromatography (HPLC) system. The amount of identified individual phenols is lower than the results obtained by the Folin–Ciocalteau method, because the F–C method also interacts with proteins, carbohydrates, amino acids, unsaturated fatty acids, and vitamins, not only with phenols [33].

Table 4 shows that p-coumaric acid accounted for the majority of phenolic compounds in this study, followed by rosmarinic acid, chlorogenic acid, and vanillic acid. Protocatechuic acid, p-hydroxybenzoic acid, and caffeic acid for Ultra-Turrax pre-treatment were found in lower quantities. Hydroxycinnamic acids, ferulic acid, coumaric acid, sinapic acid, and caffeic acid were the main phenolic compounds identified in spent grain [33,34,35,37,38]. The differences between the methodologies applied for the two extraction methods: UT is based on high-stringing rates of 10,000 to 30,000 rpm at ambient temperature in a relatively short time of 30–90 s, while US is based on the sonication of samples in an ultrasound bath at temperatures between 30 and 70 °C at an interval of 10–30 min. Some more sensitive polyphenols can be destroyed with temperature. Menses et al. and del Rio et al. reported in their studies that these differences between compounds can be attributed to the fact that the extractions were performed with different solvents, in different concentrations, which can increase the extraction of phenol compounds [29,33]. A more polar environment created from the addition of water to organic solvents can facilitate the extraction of soluble compounds in both organic compounds and/or water [43].

### 2.4. FTIR Analysis

A means for identifying functional groups in the bioliquid-to-solid ratio is FTIR spectroscopy, and the FTIR spectra are presented in Figure 5. FTIR analysis was performed to observe any alteration in the composition of the SG by assessing possible changes in functional groups after pre-treatment and compared with native SG [45].

The peaks that were identified in the spectra of the samples in the region of 3000–2850 cm^−1^ were assigned to symmetric and asymmetric stretching of C-H due to the presence of carbohydrates and lipids. The peak at 3287 cm^−1^ corresponds to OH stretching and can be attributed to cellulose, and two peaks at 2921 and 2852 cm^−1^, corresponding to stretching of C=O, respectively, to aliphatic C-H stretch. No triple-bond region was detected in the area 2000–2500 cm^−1^, resulting in no C≡C bond in spent grain. A broad band between 1730 and 1530 cm^−1^ includes C=O stretching due to amide I and II, vibrations specific to C-N stretching and N-H deformation (amide II), and C=C stretching in aromatic structures. The aromatic compounds include p-coumaric, ferulic, sinapic, and caffeic acids, which are important phenol compounds. A peak at 1629 cm^−1^ is associated with the presence of aldehydes, ketones and carboxyl groups (conjugated C=O stretch), and the peak at 1538 cm^−1^ corresponds to the vibration of C=O, which corresponds to the double-bond region. Several peaks in the fingerprint region are at 1454 cm^−1^, which represents H-C-H and O-C-H in plane bending vibration, peaks at 1239 and 1025 cm^−1^ represent C-C, C-OH, C-H ring and side group vibrations, which can be attributed to hemicellulose and lignin [46].

## 3. Materials and Methods

### 3.1. Materials

The distillery’s spent grain was kindly provided by Alexandrion Group Romania (Ploiesti, Romania). Spent grain (SG) was stored at −18 degrees Celsius and then dried at 50 degrees Celsius for 26 h to a moisture level of 5%. Furthermore, the distillery’s dried spent grain was ground and sieved for 30 min at amplitude 70 through a Retsch Vibratory Sieve Shaker AS 200 basic (Haan, Germany) to obtain spent grain flours at three fractions with different particle sizes, large (L 300–500 μm), medium (M 200–300 μm), and small (S < 200 μm) fractions. Spent grain with particle sizes of 200–300 µm was used in this study. The ground spent grain was stored in paper bags at room temperature until further use. All chemicals used in this paper were of analytical grade and were purchased from Sigma Aldrich (St. Louis, MO, USA).

### 3.2. Proximate Composition

Moisture content was determined by drying the samples in the oven (Memmert, GmbH & Co. KG, Germany) at 103 °C until a constant weight was achieved. The moisture content was defined as:MC = (**m****_0_** − **m**)/**m****_1_** ∗ 100(7)
where **m_0_** = mass of empty container (g), **m** = mass of container and sample after drying (g), and **m_1_** = mass of the sample (g).

Ash content was determined at 550 °C for 6 h in a calcination furnace. Protein content was estimated using the Kjeldhal method; the conversion factor for SG was 6.25. Total dietary fiber (TDF) content was measured according to Lee et al. [47] using a reagent kit (K-TDFR, Megazyme Int., Wicklow, Ireland). Lipids were determined using the Soxhlet method with n-hexane solvent.

### 3.3. FT-IR Analysis

Fourier transform infrared spectroscopy (FTIR) analysis using a Nicolet iS10 spectrometer from Thermo Scientific (Karlsruhe, Dieselstraβe, Germany), mounted with an attenuated total reflectance (ATR) accessory and equipped with a diamond crystal, was used for analysis. Measurements were performed in reflective absorbance mode (ATR-FTIR), at 4 cm^−1^ resolution in the range of a mid-infrared region of 650–4000 cm^−1^ with 32 scans in transmission mode. A sample of SG was placed on the ATR crystal and OMNIC software (version 32, Thermo Scientific) was used for collecting the spectra.

### 3.4. Extract Preparation

The principle of the Ultra-Turrax method is based on the homogenization of samples at high speeds of shaking at ambient temperature for a short period of time. The ultrasound-assisted method involves ultrasonication of samples in an ultrasound bath, varying temperature and time parameters, maintaining constant strength and amplitude. To our knowledge, there are no other studies related to the optimization of ultrasound-assisted extraction and Ultra-Turrax methods of polyphenols from spent grain from malt whisky.

For the comparison of ultrasound-assisted and Ultra-Turrax methods a set of determinations have been made by conventional methods. Thus, according to the method described by Meneses et al. with several modifications, a 0.875 g sample was mixed with 30 mL methanol 78% in Erlenmeyer flasks which were covered to avoid solvent loss, and maintained during 22 min in a water bath with magnetic agitation at 70 °C. The extract was centrifuged and the supernatant was collected [29]. Another conventional method used was maceration, where 0.875 g of the sample was mixed with 30 mL water and kept at ambient temperature (22 °C) for 24 h.

#### 3.4.1. Ultra-Turrax Extraction

Two pre-treatments were used for this study: ultrasound-assisted pre-treatment and Ultra-Turrax pre-treatment using a high stirring rate. The recovery of phenols from by-products is influenced by the selection of the extraction parameters, such as liquid-to-solid ratio, pH, extraction time, sample particle size, stirring rate, the temperature of extraction, or solvent concentration [33]. In the first case, methanol was used as a solvent in different concentrations (40%, 60% and 80%, *v*/*v*) with solid: liquid variations: (0.3 g, 0.6 g and 0.9 g reported solid mass to 30 mL solvent). Methanol is considered a good solvent for low molecular weight polyphenols [48]. The samples were homogenized with Ultra-Turrax homogenizer (Daigger Scientific Inc. and its affiliates, Illinois, USA) for different times (30 s, 60 s, and 90 s) at different stirring rate (10,000 rpm, 20,000 rpm, and 30,000 rpm). Samples were centrifuged at 3000 rpm for 5 min and the supernatant was collected, then they were stored in afreezer until further analysis.

#### 3.4.2. Ultrasound-Assisted Extraction

In the second case, methanol was used in variable concentrations (40%, 60% and 80% *v*/*v*), with variations of solid: liquid (0.3 g, 0.6 g and 0.9 g solid mass) reported to 30 mL solvent). The samples were sonicated in an ultrasonic bath (Elma Transsonic TI-H-15, Elma Hans Schmidbauer GmbH & Co. KG, Singen, Germany), internal dimensions: 300 × 240 × 200 mm), using a frequency of 45 kHz. After sonication, the samples were centrifuged at 4000 rpm for 15 min. The supernatant was collected, and the samples were stored in afreezer until further analysis.

### 3.5. Total Phenolic Content (TPC)

The phenolic content was determined using the Folin–Ciocalteau method as described: 0.2 mL of extract (prepared as described in Section 3.4.1 and Section 3.4.2) was mixed with 2 mL of Folin–Ciocalteau reagent, diluted 1:10, and 1.8 mL of sodium carbonate 7.5% (*w*/*v*) into a tube [49]. The mixture was left for 30 min at room temperature in the dark. The TPC was determined at 750 nm wavelength using a UV-VIS-NIR spectrophotometer (Shimadzu Corporation, Kyoto, Japan). The calibration curve of the polyphenols was performed by using gallic acid at concentrations of 10–200 mg/L with the regression coefficient R^2^ = 0.99872 and equation y = 0.00949x + 0.02950. The samples were analyzed in triplicate.

### 3.6. Flavonoid’s Content (TFC)

Flavonoids were quantified by the aluminum trichloride method as described by Spinelli et al. [50], with slight modifications. Briefly, 0.5 mL of extract was mixed with 2 mL of distilled water and 150 µL of NaNO_2_ 5% solution. Subsequently (after 6 min), 150 µL of 10% AlCl_3_ solution was added. After 6 min of reaction, 1 mL of 1 N NaOH and 1.2 mL of methanol were added to the mixture. The samples were vortexed and filtered through a 0.45 µm syringe filter (ChromafilXtra PTFE-45/25, Macherey-Nagel GmbH & Co, Düren, Germany). The samples were measured at 415 nm using a UV-VIS-NIR spectrophotometer (Schimadzu Corporation, Kyoto, Japan). Quercetin standard solutions were used for constructing the calibration curve (6.25–200 mg/L; R^2^ = 0.95946, equation y = 0.00077x + 0.00859). Total flavonoid content (TFC) was expressed as mg of quercetin equivalents (QEs) per gram of driedSG. The analyses were carried out in triplicate for each sample.

### 3.7. DPPH Assay Scavenging Activity

The 2, 2-diphenyl-1-picrylhydrazyl (DPPH) scavenging activity of the pre-treated SG was evaluated by adding 2 mL of each sample with 2 mL of DPPH solution 0.1 mM in methanol (9.85 mg DPPH dissolved in 250 mL methanol) [51]. The mixture was shakenfor 2 min and was kept at room temperature, in a dark place, for 30 min, and then the absorbance was determined at 517 nm using a UV-VIS-NIR spectrophotometer (Schimadzu Corporation Kyoto, Japan). The antioxidant capacity was measured in three repetitions for each sample, using distilled water as a blank sample.
% inhibition of DPPH = [(1 − AS/Ab)] × 100(8)
where As = absorbance of sample, Ab = absorbance of blank sample.

### 3.8. Analysis of Phenolic Compound by HPLC-DAD

Analysis was made using ahigh-performance liquid chromatography (HPLC) (Shimadzu, Kyoto, Japan) system equipped with a LC-20 AD liquid chromatograph, SIL-20A autosampler, CTO-20AC column oven and an SPD-M-20A diode array detector. The separation was carried out on a Phenomenex Kinetex^®^ 2.6 μm Biphenyl 100 Å HPLC Column, 150 × 4.6 mm, at 25 °C. The sample injection volume was 10 µL. A solvent system consisting of 0.1% acetic acid in water (solvent A) and acetonitrile (solvent B) was used with the following gradient: starting with 100% A and installing a gradient to obtain 5% B at 6.66 min, 40% B at 66.6 min and 80% B at 74 min, according to the method previously described with some modifications [52]. The solvent flow rate was 1 mL/min. Before injection, extracts were filtered through 0.45 µm pore size. Phenolic compounds were identified on the basis of the retention times of standard materials, and the quantification was achieved by the absorbance recorded in the chromatograms relative to external standards, at 280 nm for gallic acid, protocatechuic acid, vanillic acid, p-hydroxybenzoic acid, and 320 nm for chlorogenic acid, caffeic acid, p-coumaric acid, rosmarinic acid, myricetin, quercetin, luteolin and kaempferol. All standard calibration curves showed high degrees of linearity (R^2^ > 0.99).

### 3.9. Statistical Analysis

The experiment was conducted according to the Box–Behnken design, in a four-factor full factorial design. Each independent variable (time, liquid to solid ratio, concentration of solvent, and stirring rate) for Ultra-Turrax extraction and liquid-to-solid ratio, concentration of solvent, time and temperature for ultrasound extraction had at least 3 levels, as follows: for Ultra-Turrax time (30, 60 and 90 s), liquid-to-solid ratio (0.30, 0.60 and 0.90 g), concentration of methanol (40, 60 and 80% *v*/*v*) and stirring rate (10,000, 20,000 and 30,000 rpm).

For the ultrasonic treatment, the independent variables were liquid-to-solid ratio (0.3, 0.6 and 0.9 g), concentration of methanol (40, 60 and 80% *v*/*v*), time (10, 20 and 30 min) and temperature (30, 50 and 70 °C). Total phenolic content, flavonoids, and DPPH assay scavenging activity were the response of the experimental design. The full factorial design was made using Design Expert 11 (trial version, Stat-Ease Inc., Minneapolis, MN, USA). The model used to predict the output parameters was a second-order (quadratic) polynomial response surface model obtained by the Box–Behnken design. Response surface methodology (RSM) was used as an effective statistical technique for optimizing complex processes [33]. The optimized conditions of extraction were validated for the maximum phenolic content, maximum flavonoid content, and antioxidant activities using RSM.

## 4. Conclusions

Spent grain can be considered a potential available source of antioxidant compounds. Ultrasound-assisted and Ultra-Turrax (high stirring rate) pre-treatments are efficient methods to extract phenolic compounds using methanol as an organic solvent. The maximum total phenolic content found in this study was 1.76 mg GAE/g SG DW for Ultra-Turrax pre-treatment, and 2.11 GAE/g SG DW for ultrasound-assisted pre-treatment, and the maximum flavonoid content was 1.67 mg QE/g SG DW and 1.63 mg QE/g SG DW, respectively. High-performance liquid chromatography (HPLC) revealed phenolic compounds in different concentrations present in both extraction methods. The main compounds found were p-coumaric (20.4 ± 1.72 mg/100 g SG DW, and 14.0 ± 1.14 mg/100 g SG DW, respectively), rosmarinic (6.5 ± 0.96 mg/100 g SG DW, and 4.0 ± 0.76 mg/100 g SG DW) and vanillic acids (3.1 ± 0.8 mg/100 g SG DW, and 10.0 ± 1.03 mg/100 g SG DW).

Solid–liquid extraction using methanol as an organic solvent is an efficient method to extract antioxidant compounds from spent grain, due to its high polarity and high extraction yields. The obtained results emphasized the importance and the opportunities of the reuse of by-products.

Further research should be made to observe the effectiveness of these extracts in added-value food. The reuse of this by-product in food (muffins, cakes, biscuits, etc.) brings benefits both from an economic and environmental point of view, thus reducing pollution.

## Figures and Tables

**Figure 1 molecules-26-03236-f001:**
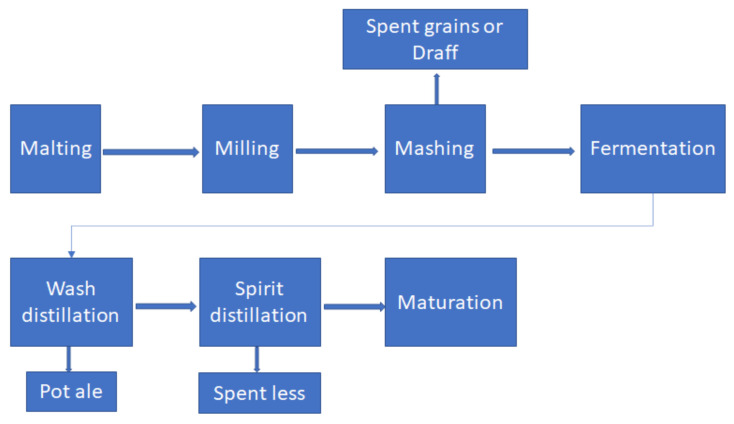
The process of obtaining whisky and the resulting by-products [13].

**Figure 2 molecules-26-03236-f002:**
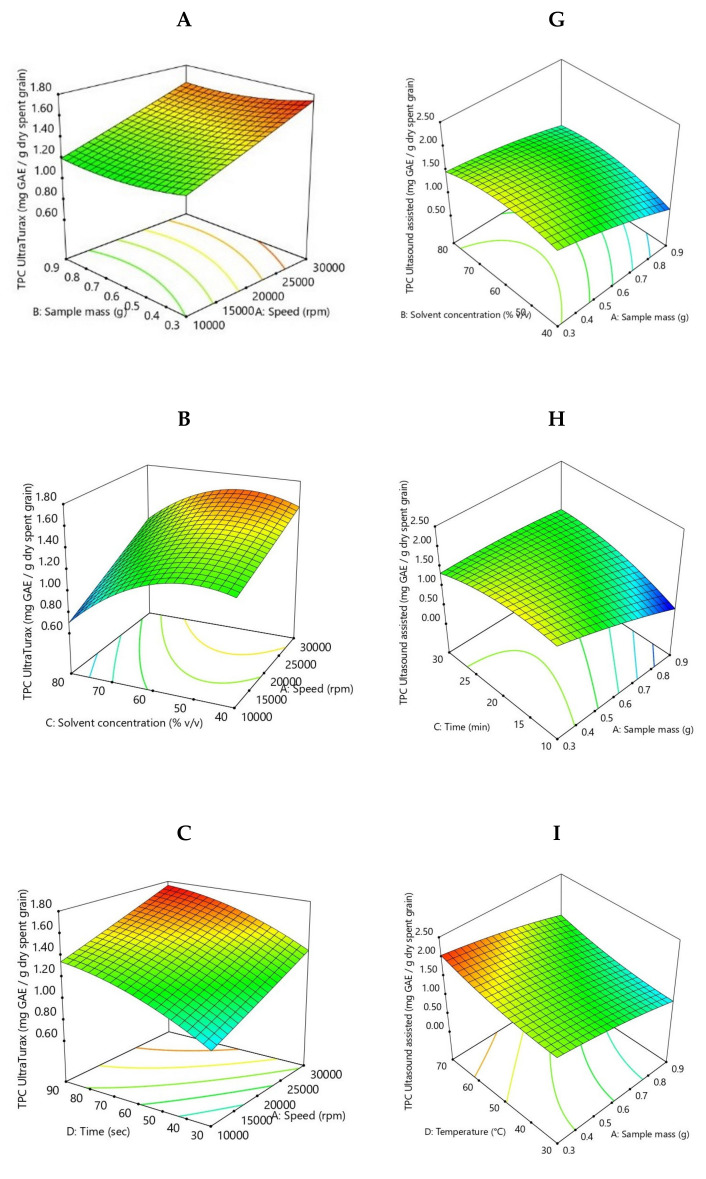
Response surface plots showing the effect of extraction parameters on phenolic compounds for SG Ultra-Turrax pre-treatment (**A**–**F**), and for ultrasound-assisted pre-treatment (**G**–**L**).

**Figure 3 molecules-26-03236-f003:**
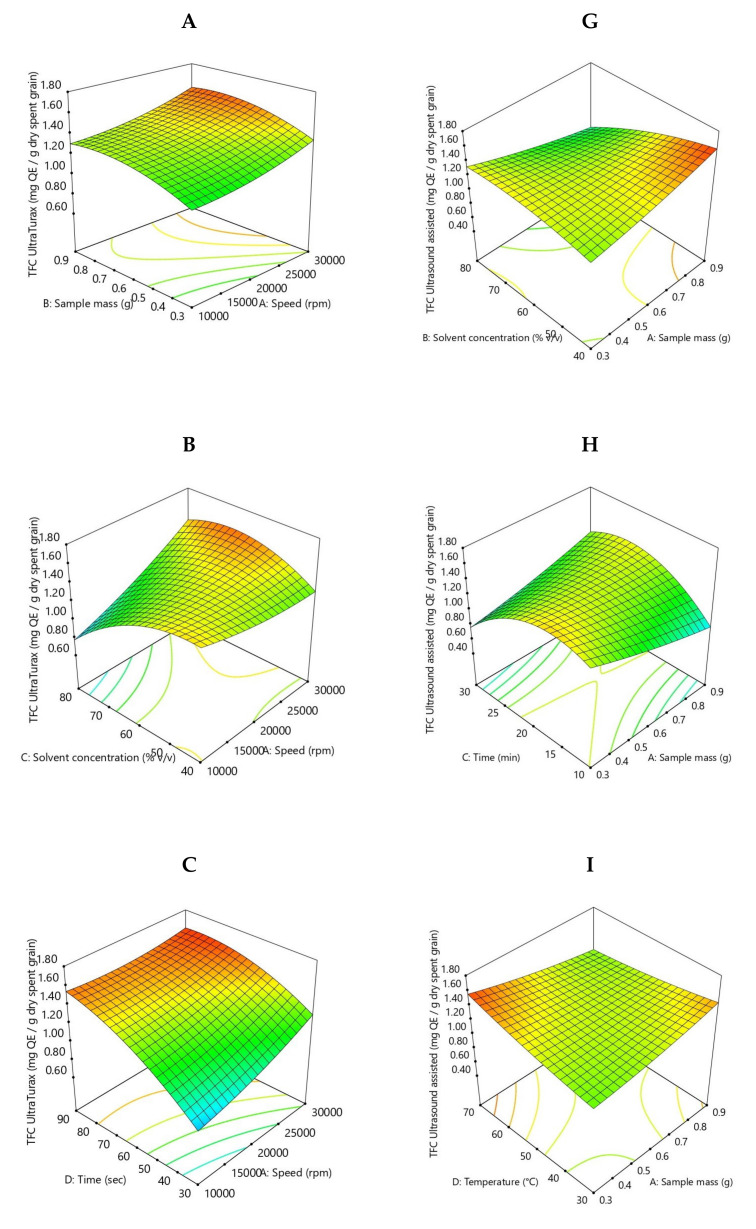
Response surface plots showing the effect of extraction parameters on flavonoid content for SG Ultra-Turrax pre-treatment (**A**–**F**), and for ultrasound-assisted pre-treatment (**G**–**L**).

**Figure 4 molecules-26-03236-f004:**
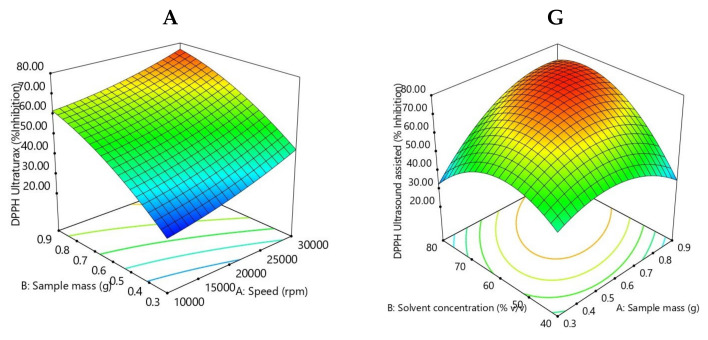
Response surface plots showing the effect of extraction parameters on antioxidant activity for SG Ultra-Turrax pre-treatment (**A**–**F**), and for ultrasound-assisted pre-treatment (**G**–**L**).

**Figure 5 molecules-26-03236-f005:**
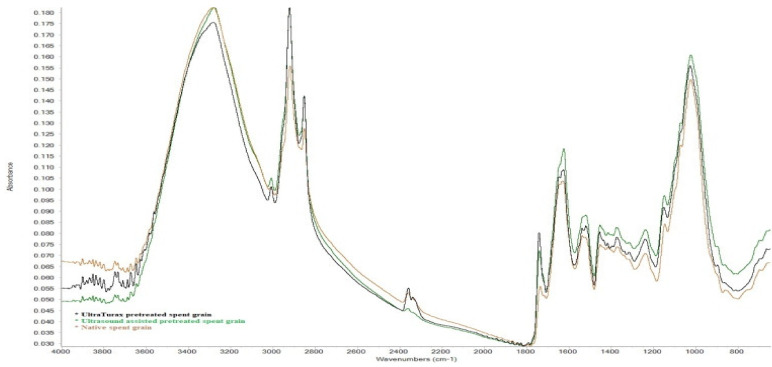
FTIR spectra of native SG and SG pretreated.

**Table 1 molecules-26-03236-t001:** Actual and coded values for full factorial experimental design.

Factor	Ultra-Turrax Pretreatment			Ultrasound-Assisted Pretreatment		
	−1	0	+1	−1	0	+1
Stirring rate (rpm) (X_1_)	10,000	20,000	30,000			
Sample mass (g) (X_2_)	0.3	0.6	0.9			
Solvent concentration (%*v*/*v*) (X_3_)	40	60	80			
Time (sec) (X_4_)	30	60	90			
Sample mass (g) (X_1_)				0.3	0.6	0.9
Solvent concentration (%*v*/*v*) (X_2_)				40	60	80
Time (sec) (X_3_)				10	20	30
Temperature (°C) (X_4_)				30	50	70

**Table 2 molecules-26-03236-t002:** ANOVA results for model fitting.

	Ultra−Turax Extraction			Ultrasound Assisted Extraction		
	TPC	TFC	DPPH	TPC	TFC	DPPH
R^2^	0.9173	0.9031	0.9477	0.8882	0.8994	0.9483
Adjusted R^2^	0.8208	0.7900	0.8868	0.7578	0.7820	0.8881
F value	9.51	7.99	15.54	6.81	7.66	15.73
*p* value	0.0002	0.0005	<0.0001	0.001	0.0006	<0.0001
Lack of Fit	0.0126	0.3498	0.0529	0.6296	0.1416	0.4432
Constant	+1.40	+1.36	+54.75	+1.44	+1.28	+74.13
X_1_	+0.22 ***	+0.12 **	+8.42 ***	−0.32 ***	−0.02	+7.03 ***
X_2_	−0.05	+0.10 *	+16.89 ***	+0.07	−0.14 **	+5.92 **
X_3_	−0.20 ***	−0.11 *	+1.86	+0.12 *	−0.02	+9.13 ***
X_4_	+0.22 ***	+0.28 ***	+6.45 ***	+0.25 ***	+0.04	+7.99 ***
X_1_X_2_	−0.01	+0.002	−0.97	+0.11	−0.20 *	+12.19***
X_1_X_3_	+0.05	+0.21	+2.38	+0.27*	+0.23 **	+3.59
X_1_X_4_	−0.01	−0.09**	−1.58	−0.001	−0.15 *	−1.38
X_2_X_3_	+0.03	+0.08	+0.09	−0.19	+0.22 *	−2.83
X_2_X_4_	−0.04	−0.06	−0.31	−0.02	−0.03	−1.05
X_3_X_4_	−0.004	−0.007	+1.27	−0.16	−0.47 ***	−9.22 **
X_1_^2^	+0.004	+0.05	+2.08	−0.06	+0.03	−11.77 ***
X_2_^2^	+0.05	−0.09	−4.41	−0.18 *	−0.07	−16.27 ***
X_3_^2^	−0.24 **	−0.19 **	+2.90	−0.19 *	−0.32 ***	−10.06 ***
X_4_^2^	−0.08	−0.12	−0.51	+0.10	+0.03	−4.83 *

**TPC**: total polyphenol content, **TFC**: total flavonoid content, **DPPH**: scavenging 2, 2-diphenyl−1-picrylhydrazyl free radical activity, ********p* < 0.001; ******
*p* < 0.01; *****
*p* < 0.05.

**Table 3 molecules-26-03236-t003:** Predicted and verified optimal sample characteristics.

Factor	Ultra-Turrax Pretreatment		Ultrasound-Assisted Pretreatment	
	Predicted Value	Verified Value	Predicted Value	Verified Value
X_1_	29,999.9 rpm	30,000 rpm	0.875 g	0.875 g
X_2_	0.849 g	0.849 g	77.7%	78%
X_3_	54.0%	54%	21.7 min	22 min
X_4_	89.9 s	90 s	68.8 °C	70 °C
TPC (mg GAE/g dry spent grain)	1.40 ± 0.13	1.37 ± 0.11	1.44 ± 0.19	1.42 ± 0.15
TFC (mg QE/g dry spent grain)	1.36 ± 0.13	1.34 ± 0.15	1.29 ± 0.15	1.25 ± 0.11
DPPH (%Inhibition)	54.75 ± 4.85	54.77 ± 3.75	74.13 ± 5.03	74.20 ± 4.39

**TPC**: total polyphenol content, **TFC**: total flavonoid content, **DPPH**: scavenging 2, 2-diphenyl-1-picrylhydrazyl free radical activity.

**Table 4 molecules-26-03236-t004:** Phenolic profile of spent grain obtained in optimal conditions using the HPLC-DAD method.

Compound	Molecular Weight	Wavelength (nm)	Retention Time (min)	Ultra-Turrax Phenolic Content (mg/100g dw)	Ultrasound-Assisted Phenolic Content (mg/100g dw)
Protocatechuic acid	154.12	280 nm	13.756	0.7 ± 0.05	N.D.
p-hydroxybenzoic acid	138.12	280 nm	17.876	1.1 ± 0.06	N.D.
Vanillic acid	168.14	280 nm	22.627	3.1 ± 0.8	10.0 ± 1.03
Caffeic acid	180.159	320 nm	20.557	0.7 ± 0.03	N.D.
Chlorogenic acid	354.31	320 nm	21.971	5.4 ± 1.1	N.D.
p-coumaric acid	164.047	320 nm	27.503	20.4 ± 1.72	14.0 ± 1.14
Rosmarinic acid	360.31	320 nm	30.989	6.5 ± 0.96	4.0 ± 0.76

## Data Availability

The data presented in this study are available in this article.

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
