# Peer review of "Spent Grain from Malt Whisky: Assessment of the Phenolic Compounds"

_molecules, 2021, doi:10.3390/molecules26113236_

Round 1
Reviewer 1 Report
Point 1: general comments: - the manuscript should undergo English editing, even when I´m not native English speaking some grammatical errors are evident.
Point 2: A comparison with conventional solvents should be carried out under the same experimental conditions.
Point 3: Table 1 is not proper address. There are no unit for this sample and it does specify what other samples refers to.
Point 4: Authors should explain why they determine the mineral content when they are interested in the phenolic compounds (as describe in the title) Furthermore there es no deep discussion of those results.
Point 5: Regarding Table 4. How does the author explain why the US treatment have less identified compound when it has the higher values? Indicating that is relate to solvent since it is the same (line 314), and the overall ratio between sample and solvent seem similar.
Point 6: Why only one antioxidant assay was performed? To better asses the antioxidant activity a set of assays should be performed.
Point 7: Why author did not have HPLC-MS/MS to identify and quantify the phenolic compounds, according to literature it should be other compounds present and a more selective tool should be employed.
Point 8: Why performed FTIR? The conclusion is related to cellulose-lignin type compounds and not related to phenolic compounds as stated in the title.
Point : General corrections, some examples:
Line 184 Improve caption of Figure 2
Line 384 Correct flame components
Line 409 correct cm-1
Author Response
Dear Referee,
Thank you very much for the recommendations to publication our paper. We appreciate very much your suggestions. We hope that we provide all the answers to the reviewer’s comments.
"Spent Grain from Malt Whisky: Assessment of the Phenolic Compounds",
In the original document we use red colour for added text. We look forward to hear from you soon.
Sincerely yours,
Adriana Dabija and Ancuța Chetrariu
Point 1: general comments: - the manuscript should undergo English editing, even when I´m not native English speaking some grammatical errors are evident.
First of all, we would like to thank the referee for the close reading and for all the given comments suitable for improving the manuscript. The entire manuscript was reviewed by an English teacher.
Point 2: A comparison with conventional solvents should be carried out under the same experimental conditions.
We want to thank to referee for her/his suggestion. Thanks to your proposal we made an analogy with the data presented in the literature.
Point 3: Table 1 is not proper address. There are no unit for this sample and it does specify what other samples refers to.
We would like to thank to the referee for her/his suggestions in order to improve the quality of the manuscript. We revised the manuscript according to referee suggestion: the data was entered in the manuscript and compared with the data from the literature.
Point 4: Authors should explain why they determine the mineral content when they are interested in the phenolic compounds (as describe in the title) Furthermore there es no deep discussion of those results.
We want to thank to referee for her/his suggestion. We agree with her/his remark that mineral are not the subject of this study. Therefore, we agree with her/him point of view.
Point 5: Regarding Table 4. How does the author explain why the US treatment have less identified compound when it has the higher values? Indicating that is relate to solvent since it is the same (line 314), and the overall ratio between sample and solvent seem similar.
We would like to thank to the referee for her/his remarks. Individual phenolic compounds were identified using the optimals obtained for both pre-treatments used. A lower concentration of solvent using the same solvent ratio:liquid requires more water, which creates a more polar environment that can facilitate the extraction of compounds (54,005% methanol aqueous mixture for Ultra-Turax vs. 77,782% methanol aqueous mixture for Ultrasound assisted pre-treatment). The differences between the methodologies applied for the two extraction methods: UT is based on high-stringing rates of 10000 to 30000 rpm at ambient temperature in a relatively short time of 30-90 seconds, while US is based on the sonication of samples in an ultrasound bath at temperatures between 30-70° Celsius in an interval of 10-30 minutes. Some more sensitive polyphenols can be destroyed with temperature.
Point 6: Why only one antioxidant assay was performed? To better asses the antioxidant activity a set of assays should be performed.
Thanks for the suggestions, you're right, indeed a wider range of tests would have a better applicability. For now, we don't have the reagents for the other tests or the samples extracted. I'd like to thank you for your suggestion once again and I'll take these into account for future research.
Point 7: Why author did not have HPLC-MS/MS to identify and quantify the phenolic compounds, according to literature it should be other compounds present and a more selective tool should be employed.
We would like to thank to the referee for her/his remarks. We are considering purchasing HPLC-MS/MS equipment in the future, but for the moment only this equipment (HPLC-DAD) we have in the laboratory of our faculty. Clearly HPLC-MS/MS is more selective and more indicated. Below is the link with the equipment available to our faculty.
https://erris.gov.ro/Instrumental-Analytical-Cent
Point 8: Why performed FTIR? The conclusion is related to cellulose-lignin type compounds and not related to phenolic compounds as stated in the title.
We would like to thank to the referee for her/his remarks. FTIR analysis highlighted the changes in the chemical composition in the both pretreated spent grain as compared to the untreated one. We’ve updated the manuscript with regard to phenolic compounds.
Point : General corrections, some examples:
Line 184 Improve caption of Figure 2
We would like to thank to the referee for her/his remarks. We made the changes according to the referee suggestions.
Line 384 Correct flame components
We would like to thank to the referee for her/his remarks. We made the changes according to the referee suggestions.
Line 409 correct cm-1
We would like to thank to the referee for her/his remarks. We made the changes according to the referee suggestions.

Reviewer 2 Report
The authors optimized the extraction of antioxidant phenolic compounds from spent grain by two methods in this study. In my opinion, the manuscript has the following shortcomings to be improved.
- Absract needs spicific results.
- Introduction fails to introduce the progress of this field. Most of the information is irrelevant with the content of this study.
- More experiments about the extracts characteristics are needed to support the related conclusions. What's the deferences of the two methods? and the mechanism of them should be discussed.
Author Response
Dear Referee,
Thank you very much for the recommendations to publication our paper. We appreciate very much your suggestions. We hope that we provide all the answers to the reviewer’s comments.
"Spent Grain from Malt Whisky: Assessment of the Phenolic Compounds",
In the original document we use yellow colour for added text. We look forward to hear from you soon.
Sincerely yours,
Adriana Dabija and Ancuța Chetrariu
The authors optimized the extraction of antioxidant phenolic compounds from spent grain by two methods in this study. In my opinion, the manuscript has the following shortcomings to be improved.
- Absract needs spicific results.
We would like to thank to the referee for her/his remarks. We added specific results in the abstract side according to the evaluator's suggestions.
- Introduction fails to introduce the progress of this field. Most of the information is irrelevant with the content of this study.
Thank you for careful reading of our manuscript. We agree with the evaluator's suggestions and have improved the introduction part with more information related to manuscript content (we excluded irrelevant information).
- More experiments about the extracts characteristics are needed to support the related conclusions. What's the deferences of the two methods? and the mechanism of them should be discussed.
We would like to thank to the referee for her/his remarks. The principle of the UltraTurax method is based on the homogenization of samples at high speeds of shaking at ambient temperature for a short period of time. Ultrasound assisted method involves ultrasonication of samples on an ultrasound bath, varying temperature and time parameters, maintaining constant strength and amplitude. From our knowledge, this comparison has not been made between these two pre-treatments for spent grain.

Reviewer 3 Report
In this manuscript the phenolic compounds from spent grain from Malt Whisky were extracted by using two extraction methods: ultrasound assisted method and ultraTurax method (high stirring rate).
The manuscript needs major revision before being considered for pubication in Molecules. Below you will find some important points you need to pay attention:
- Table 3 should be completed in terms of presenting the data of the ANOVA for each one of the factors, their interactions and the lack of fit of the models for each one of the response variables evaluated.
- All the Figues should be re-edited, since it is no posible to read either the levels of the axis or the legends.
- The results described in lines 188 to 194 are not clear. Please rewrite and explian better if the materials are different.
- The optimal conditions found for the two extraction methods should be validated experimentally in order to verify if the predicted values obtained using the models, correspond to the experimental ones.
- The results presented in Table 4 should be explained. What is ths reason of the different phenolic profile as a function of the extraction method? In lines 314 to 316 you said that the difference is attributed to different solvents, but the solvent was the same in both extractions.
Author Response
Dear Referee,
Thank you very much for the recommendations to publication our paper. We appreciate very much your suggestions. We hope that we provide all the answers to the reviewer’s comments.
"Spent Grain from Malt Whisky: Assessment of the Phenolic Compounds",
In the original document we use blue colour for added text. We look forward to hear from you soon.
Sincerely yours,
Adriana Dabija and Ancuța Chetrariu
In this manuscript the phenolic compounds from spent grain from Malt Whisky were extracted by using two extraction methods: ultrasound assisted method and ultraTurax method (high stirring rate).
The manuscript needs major revision before being considered for pubication in Molecules. Below you will find some important points you need to pay attention:
- Table 3 should be completed in terms of presenting the data of the ANOVA for each one of the factors, their interactions and the lack of fit of the models for each one of the response variables evaluated.
We would like to thank to the referee for her/his remarks. We've improved the data in the table with your remarks.
- All the Figues should be re-edited, since it is no posible to read either the levels of the axis or the legends.
We would like to thank to the referee for her/his remarks. We made the changes according to the referee suggestions, we re-edited the figures and improved them.
- The results described in lines 188 to 194 are not clear. Please rewrite and explian better if the materials are different.
We would like to thank to the referee for her/his suggestions in order to improve the quality of the manuscript. We revised the manuscript according to referee suggestion. Spent grain was subjected at two pretreatments: UltraTurax homogenization at high shaking speeds in a relatively short time of seconds and sonications per ultrasonic bath, varying temperature and time. The solvent used in both situations was an aqueous mixture of methanol.
- The optimal conditions found for the two extraction methods should be validated experimentally in order to verify if the predicted values obtained using the models, correspond to the experimental ones.
We would like to thank to the referee for her/his remarks. We added a table with Predicted values and Verified values for both UltraTurax and Ultrasound assisted methods.
- The results presented in Table 4 should be explained. What is ths reason of the different phenolic profile as a function of the extraction method? In lines 314 to 316 you said that the difference is attributed to different solvents, but the solvent was the same in both extractions.
We would like to thank to the referee for her/his remarks. Individual phenolic compounds were identified using the optimals obtained for both pre-treatments used. A lower concentration of solvent using the same solvent ratio:liquid requires more water, which creates a more polar environment that can facilitate the extraction of compounds (54,005% methanol aqueous mixture for Ultra-Turax vs. 77,782% methanol aqueous mixture for Ultrasound assisted pre-treatment). The differences between the methodologies applied for the two extraction methods: UT is based on high-stringing rates of 10000 to 30000 rpm at ambient temperature in a relatively short time of 30-90 seconds, while US is based on the sonication of samples in an ultrasound bath at temperatures between 30-70° Celsius in an interval of 10-30 minutes. Some more sensitive polyphenols can be destroyed with temperature.

Round 2
Reviewer 1 Report
- Point 2 (from previous questions) was not completely addressed. A comparison with conventional methods should be carried out under the same experimental conditions and not only reported as a references
- Point 6 (from previous questions) was not completely addressed. In order to concluded about the antioxidant capacity more assay should be performed.
- Point 7 (from previous questions) was not completely addressed, since the HPLC-DAD system does not give enough information about the polyphenols profile.
Author Response
- Point 2 (from previous questions) was not completely addressed. A comparison with conventional methods should be carried out under the same experimental conditions and not only reported as a references
We performed a comparison of ultrasound assisted and ultraTurax methods with conventional methods by maceration at ambient temperature and magnetic shaking on the water bath at 70°C. The results were slightly lower than the methanolic extractions resulting from ultrasonic and shaking at very high speeds, results influenced by the amount of lignin in SG that is connected to its cell wall polysaccharides by phenolic acids. Thank you for your attention to our manuscript and for your guidance.
- Point 6 (from previous questions) was not completely addressed. In order to concluded about the antioxidant capacity more assay should be performed.
2,2-diphenyl-1-picrylhydrazyl (DPPH) radical scavenging method is extensively used to determine the antioxidant potential of various plant extracts and natural products. Thank you for your attention to our manuscript and for your guidance.
- Point 7 (from previous questions) was not completely addressed, since the HPLC-DAD system does not give enough information about the polyphenols profile.
To determine individual polyphenols we used a standards kit to which we identified the retention time and according to those standards we identified individual phenolic compounds. Indeed, there are several phenolic compounds in the extract, but those in the manuscript have been identified on the basis of this mix of standards. Gallic acid, p-hydroxybenzoic acid, protocatechuic acid, vanillic acid, caffeic acid, chlorogenic acid, p-coumaric acid, rosmarinic acid, mircitin, luteolin, quercitin, and kaempferol were used for standards. The chromatograms were monitored at 280 nm (hydroxybenzoic acids) and 320 nm (hydroxycinnamic acids).
Stefanello et al., 2018, Ideia et al., 2020, Ikram et al., 2019, Alonso-Riano et al., 2020, and Gupta et al., 2013, used for their study to identify the individual phenolic compounds from spent grain using HPLC-DAD also. Being used in many research studies, it is considered to be a strong and effective method for determining individual phenolic compounds. Thank you for your attention to our manuscript and for your guidance.

Reviewer 2 Report
The manuscript has been improved. The authors should make revisions as follows:
- The introduction is too lengthy, too much review of the history of whisky development, and the introduction of by-products should be simplified, because these contents have little to do with this study.
- In practical application, the optimized conditions should be properly adjusted to facilitate the control of operation. For example, 0.875g sample, 77.7% solvent concentration, 21.7min, 68.8min ° C. 299999.9rpm, how to accurately control these parameters? After adjustment, please carry out the verification test and compare with the pridicted values.
- There are still some mistakes in grammar and writing format. Please check and correct them carefully.
Author Response
Dear Reviewer,
Thank you very much for the recommendations to publication our paper. We appreciate very much your suggestions. We hope that we provide all the answers to the reviewer’s comments.
"Spent Grain from Malt Whisky: Assessment of the Phenolic Compounds",
We include all the answers in the manuscript.
We look forward to hear from you soon.
Sincerely yours,
Adriana Dabija and Ancuța Chetrariu
1. The introduction is too lengthy, too much review of the history of whisky development, and the introduction of by-products should be simplified, because these contents have little to do with this study.
We want to thank to referee for her/his suggestion. We agree with her/his remark that the introduction is too lengthy. Therefore, we agree with her/him point of view and we modified the introduction.
- In practical application, the optimized conditions should be properly adjusted to facilitate the control of operation. For example, 0.875g sample, 77.7% solvent concentration, 21.7min, 68.8min ° C. 299999.9rpm, how to accurately control these parameters? After adjustment, please carry out the verification test and compare with the pridicted values.
We've adjusted the parameters so that they're as close as possible to the predicted values. We carried out the checks with the adjusted values right from the beginning, but we noted the values predicted by Design Expert. Thank you for your attention to our manuscript and for your guidance.
- There are still some mistakes in grammar and writing format. Please check and correct them carefully.
We would like to thank the referee for the close reading and for all the given comments suitable for improving the manuscript. The entire manuscript was reviewed by an English teacher (The English language correction was highlighted in green).
